# Effects of a Cognitive-Behavioral Therapy Intervention on the Rowers of the Junior Spain National Team

**DOI:** 10.3390/healthcare10122357

**Published:** 2022-11-24

**Authors:** Manuel Isorna-Folgar, Raquel Leirós-Rodríguez, Santiago López-Roel, José L. García-Soidán

**Affiliations:** 1Faculty of Education and Sport Sciences, Universidade de Vigo, 32004 Ourense, Spain; 2SALBIS Research Group, Faculty of Health Sciences, Universidad de León, 24401 Ponferrada, Spain; 3Spanish Rowing Federation, 28080 Madrid, Spain

**Keywords:** water sports, cognitive-behavioral therapy, psychological techniques, motivation

## Abstract

Cognitive-behavioral therapy has been implemented in the training plans of many athletes and sports teams, obtaining very good results for the improvement of mental skills. These effects are achieved through improvements in cognitive skills such as motivation, mental concentration, and self-confidence, all within an appropriate context of specific physical, technical, and tactical training. However, there are few studies that consider the analysis of performance from a psychological point of view from a gender perspective. The objective of this study was to evaluate a mental training intervention (cognitive-behavioral therapy) in youth rowers in preparation for their participation in the Junior European Championship. A quasi-experimental study was carried out with the complete team of the Spanish youth rowing team (*n* = 16). The setting where the intervention took place was during the team’s stay prior to the European Rowing Junior Championships. Psychological variables were assessed using the Psychological Characteristics Related to Sport Performance Questionnaire. The final assessment was carried out the week before the championship (after finishing the 10th week of intervention). After the intervention, improvements were identified in the Stress Control and Influence of Performance Evaluation subscales in the total sample. However, in the analysis separated by sex, only women improved on the Influence of Performance Evaluation subscale. Lastly, the linear regression analysis adjusted for the sex variable was only significant with the age variable (with a negative effect). This intervention was found to be effective in improving only some of the psychological components related to athletic performance (Stress Control and Influence of Performance Evaluation). These improvements were not related to better results in the European Rowing Junior Championship. These results should be taken into account because they provide evidence for the efficacy of psychological interventions in the field of grassroots sport.

## 1. Introduction

The training process of athletes, from beginners to elites, depends on biological, psychological, perceptual-cognitive, and social factors [1]. Psychological components can be decisive when facing both training and competition, regardless of the sport practiced [2]. In this sense, psychological interventions are a useful tool to increase psychological well-being and sports performance [3]. These effects are achieved through improvements in cognitive skills such as motivation, mental concentration, and self-confidence [4,5], all of which are appropriate in the context of specific physical, technical, and tactical training [1].

Psychological training directly affects athletic development through the learning and improvement of skills and appropriate coping strategies [3] and the treatment of psychological well-being, which is especially relevant in young sportsmen and women due to the importance of this vital stage in individual maturation [6]. Thus, it is essential for psychologists to carry out individualized intervention with athletes in their reflection and decision-making processes. At the same time, these must be useful and applicable to the different sports and extra-sports situations specific to their discipline [7,8].

Psychological interventions in sports are of great importance today. They increase psychological well-being and optimize sports performance. In addition, the training and/or teaching of psychological strategies improves some very important psychological skills in sports, such as mental concentration, motivation, level of arousal, etc., which are so important in sports [9,10]. At the same time, high-level athletes and participants in international competitions carry out numerous tough training sessions. This sometimes leads to possible physical and mental overload and fatigue that needs to be managed so that it does not negatively affect their health and performance [11,12].

Psychological training is a preparation program composed of different techniques that provide the athlete with the learning, maintenance, and improvement of motor and cognitive skills [10,11,12]. Palicio et al. [13], in a review of the different psychological intervention techniques and programs to optimize and improve sports performance, highlighted the most effective interventions based on mindfulness, acceptance, and commitment therapy, stress inoculation therapy, emotional freedom technique, psychological skills training, and cognitive-behavioral therapy (CBT).

Psychological skills in themselves do not produce an athlete’s performance above his or her potential. However, they can help achieve, together with physical, tactical, and technical training, that the athlete reaches a level of performance as close as possible to his or her maximum potential [14,15]. CBT has been implemented in the training plans of many athletes and sports teams, obtaining very good results for the improvement of mental skills [13,16,17,18] and emotional control [19]. As Nieto and Olmedilla [20] point out, “all general aspects of sports training can benefit from psychological intervention, thus optimizing the work methodology of coaches and athletes”. CBT has gained importance to stimulate the psychological background of athletes and has been included in the work methodology of teams and athletes [14,21,22,23,24,25,26]. However, there are few studies that consider the analysis of performance from a psychological point of view from a gender perspective [27]. Knowing whether there are differences between male and female athletes and what these differences are and in what aspects they are manifested with questions that could help improve the work of sports coaches and trainers. Thus, assessing psychological capacities is the first step to a correct psychological intervention and improving the athlete’s performance (essential for both professional players and young footballers in training) [8].

In novice athletes, promoting these cognitive skills is of special relevance to increase adherence to sports practice, avoid the wear and tear of defeats, and manage the stress of competitions [28]. These phenomena must be prevented in order to avoid their associated consequences, such as loss of attention, increased anxiety, or difficulty in mental concentration [29]. Before and during competitions, stress has been linked to motivation and increased attentional resources, as long as the stress level is not excessive and is combined with problem-focused coping styles [30].

Rowing, in its modality of a mobile bench, has been an Olympic sport since the very creation of the Olympic Games of the Modern Era, in 1896, with 11 official modalities in the World Rowing Federation (FISA). The female category was introduced at the 1974 World Championships, and at the 1976 Olympic Games. The first championships with the participation of light rowers (average weight under 70 kg) took place in 1975, and with light rowers (average weight under 57 kg) in 1983 [31].

Interventions based on cognitive-behavioral therapy are based on the thoughts, behaviors, and feelings of athletes interacting with each other in the face of environmental stimuli. The nature and weighting that athletes make of these three elements directly influence their sports performance, and vice versa. That is, thoughts, emotions, and behaviors are influenced by skill development. Therefore, all of them should be considered objects of training to optimize sports performance [32].

Thus, it was considered necessary to carry out this research in order to present an intervention program based on CBT and find out its possible effectiveness regarding the improvement of psychological skills related to sports performance and possible gender differences.

## 2. Materials and Methods

### 2.1. Study Design and Sample

A quasi-experimental study was carried out with a convenience sample consisting of the complete team of the Spanish youth rowing team (*n* = 16). The setting where the intervention took place was during the mental concentration prior to the European Rowing Junior Championships. The athletes were accommodated on a permanent camp basis at the training site for eleven weeks prior to the championship. All athletes invited to the competition (*n* = 27) were informed of the possibility of participating in this research on the day of their arrival at their homestay during the competition.

A total of 27 rowers started the team stay, of whom 16 were eventually selected to participate in the championship. Of the 11 athletes who were candidates for participation, six refused to participate, and the other five were unable to provide informed consent to participate, signed by their parents or guardians, before the start of the intervention.

### 2.2. CBT Intervention

A mental training program was carried out with the following objectives: (a) to learn to manage their level of arousal; (b) to train their ability to concentrate; (c) to optimize their self-confidence; (d) to learn to compete by improving their racing routines; and (e) to integrate mental training into their day-to-day lives.

The athletes worked in small groups of four. In addition, on the days that the psychologist attended the mental concentration, he was also present in the training sessions, with the aim of giving them feedback on the strategies they had acquired. The psychologist was present in various phases throughout the mental concentration, and there was also constant online monitoring. Each week, a sheet of tasks to be executed and a diary were provided to verify compliance with the guidelines.

In the first session, the psychologist explained to the athletes what the intervention consisted of, what they were going to work on, and what was expected of them. In addition, he explained concepts such as what sport psychology is, what a sport psychologist does, and other basic notions [33]. Afterward, there was a question-and-answer session for them to comment or ask questions, and the participants completed the Psychological Characteristics Related to Sport Performance Questionnaire (CPRD).

The 2nd and 3rd sessions dealt with mental training, motivation, mental concentration, and arousal by first explaining key knowledge such as what each of these constructs are, what types there are, and how they can be maintained or even increased [34].

In the 4th session, the psychologist developed the basic notions of goal setting [22] and gave each rower two forms, one on goals for the next competition and another on planning goals for the remainder of the season, with the intention that they would fill them in. Afterward, the group corrected how they had expressed them and whether they were measurable and feasible. The objectives for the season should always be kept in mind for the rest of the season, and before each competition, they should set two objectives to be achieved during the season. The format of the self-recorded goals for the competition was that they were to set two goals (performance, not result) before the competition, and at the end of the competition, they were to describe whether or not they had managed to achieve them and the factors that had helped or limited the achievement of the goal. As for the self-recording of long-term goals, they were to propose goals to be achieved at the end of the season and add short- and medium-term goals that would help in the achievement of these final goals.

The 5th and 6th sessions were aimed at working on attention–mental concentration: what attention and mental concentration are, what types of attention there are, and how attention can fluctuate in training and competition [35].

In the 7th session, the participants worked on the visualization technique to improve this attention–mental concentration, explaining its benefits and how to carry it out successfully [36].

The 8th and 9th sessions dealt with the level of arousal, explaining the concepts of positive and negative arousal and how they influence competition and training [37]. In addition, session 8 worked on Jacobson’s Progressive Relaxation Technique, which focused on reducing the level of arousal, how to practice it, and what benefits it brings to the athletes [38].

Finally, in the 10th session (which was established as a closing session), the psychologist made a brief summary of everything that had developed in the previous sessions. Afterward, he opened a round of questions in case there were any doubts, and the participants filled out the CPRD again.

### 2.3. Study Variables

The initial assessment was performed on the second day of the mental concentration after the explanation of the intervention was carried out. The final assessment was carried out the week before the championship (after finishing the 10th week of intervention).

Psychological variables were assessed using the CPRD [39], based on the Psychological Skills Inventory for Sports (PSIS) [40]. The questionnaire consists of 55 items graded on a 5-option Likert scale (from totally disagree to totally agree). It also includes the response option “I do not understand” to avoid missing answers. CPRD includes five subscales: Stress Control (SC), Influence of Performance Evaluation (IPE), Motivation (M), Team Cohesion (TC), and Mental Skills (MS), showing acceptable values of internal consistency for the total scale (α = 0.85) and for most of the subscales (α_SC_ = 0.88; α_IPE_ = 0.72; α_M_ = 0.67; α_TC_ = 0.78; and α_MS_ = 0.34) [41].

In addition, the following study variables were taken into account: gender, age, years of experience in the rowing discipline, and the position achieved on the podium in the European Championships (all research participants were participants in at least one individual test, and their best-achieved position was used).

### 2.4. Statistical Analysis

For the descriptive statistics, the mean was used as a measure of central tendency and the standard deviation as a measure of dispersion. The Levene’s test was used to demonstrate the homogeneity of the variance for all the variables from all the tests. To determine the normality of the variables, the Shapiro-Wilk test was performed, due to the sample size (N less than 30). To test the hypothesis of equality of means, a single-factor ANOVA was used for the variables for which the hypothesis of normal distribution was accepted. The Fisher statistic was applied due to the fact that the *p*-value was higher than 0.05 for the homogeneity of the variances in Levene’s test. To compare the groups one by one, the Tukey’s post hoc test was used.

To determine whether there was a correlation between the quantitative variables of this study, the Pearson and Spearman coefficients were used according to the type of distribution assumed for each of the variables.

We used linear regression models with the position in the European Championship as the dependent variable and age and the CPRD subscales as independent variables, with adjustments for years of experience in the rowing practice. To evaluate the fit in the linear regression models, the R^2^ statistic was used. The criteria used to evaluate the adjustment values higher than R^2^ > 0.25 were used when they were significant.

All statistical techniques were applied with STATA for MAC (version 12) and with the significance level set at *p* < 0.05.

## 3. Results

Sixteen rowers were included, of whom seven (43.8%) were women. Initially, no differences were found between genders in their age, years of rowing experience, or the position they reached in the championship (Table 1).

Before the intervention, it was identified that women athletes had significantly higher scores on the SC subscale (*p* = 0.01; d = −1.34).

After the intervention, improvement was identified in the SC and IPE subscales in the total sample (*p* < 0.01) (Figure 1). However, in the analysis separated by sex, women athletes only improved on the IPE subscale (*p* = 0.04) (Figure 1).

The comparison between genders in the last evaluation maintained the pre-existing differences in the SC scale (*p* = 0.01; d = −1.35), and they were also found in the IPE scale (*p* = 0.007; d = −0.57).

The correlation analysis between the variables age and years of rowing experience was not significant for any of the CPRD subscales. Among the CPRD subscales, significant correlations were identified for the MS subscale with the M and TC subscales (r = 0.6; *p* < 0.02 in both cases) and between the SC and IPE subscales (r = 0.8; *p* = 0.0006).

Lastly, the linear regression analysis adjusted for the sex variable was only significant with the age variable (Table 2), showing a negative effect on the position achieved in the European Championship (for each year, the variable put is multiplied by −2.58).

## 4. Discussion

The objective of this research was to evaluate a mental training intervention (cognitive-behavioral therapy) in youth athletes in preparation for their participation in the Junior European Championship. This intervention was found to be effective in only improving some of the psychological components related to athletic performance.

The results of this work are in line with what has been found in the scientific literature in recent years [3,29], highlighting the great importance of psychological intervention in sports, providing greater psychological well-being, and increased sports performance [3,29,42]. Furthermore, it is also in line with those studies that have shown that, through psychological training, psychological variables such as motivation, concentration, self-confidence, or the level of arousal [10,43], or the resources and acquisition of psychological skills to better manage their sports practice, have improved [9,44].

Specifically, the intervention managed to improve the SC and IPE components, which represent the improvement of stress management in general and that related to performance evaluation. These improvements are consistent with those obtained by previous similar programs focused on visualization, goal setting, and breathing control techniques [29,44,45]. As a specific finding, it was identified that both factors were strongly correlated in the two evaluation moments carried out in this study. Therefore, the therapeutic approach of both must be common and differentiated from the other three subscales analyzed. Their association is congruent given that both subscales refer to aspects related to stress, a conditioning factor of psychological and physical health, due to its influence on the rehabilitation and prevention of sports injuries [46,47].

Stress management has proven its usefulness and effectiveness in the field of athlete health (both physical and psychological), even in terms of physical rehabilitation or the prevention of sports injuries [48,49,50]. Moreover, learning in the sports field can be closely related to learning life skills [51]. Therefore, stress management could be a very important application tool from the sport context to the life context (daily life) of the rower [44]. In terms of psychological well-being, mental health has been considered a very important resource for athletes in relation to their performance and professional development [52].

In relation to gender, the fact that female athletes did not improve the SC component could be due to the fact that they already started from a high level in this skill. Furthermore, in all subscales (except for MS), they presented higher results than men before and after the intervention. The greater development of psychological skills in female athletes may be due to their tendency to present higher levels of self-compassion, which, in turn, favors positive self-evaluations and reduces the development of feelings of anguish (shame, irritability, sadness, nervousness, etc.) [53,54]. Therefore, the intervention also improved men’s SC skills, although it failed to compensate for pre-existing differences with female athletes.

Among the variables analyzed, it was identified that the higher the MS, the higher the M and TC. This interrelationship is consistent with the fact that the applied intervention failed to improve any of them individually. The lack of improvement in these components may be due to the individualization of the guidelines given to each athlete in this research, which is a positive aspect from the therapeutic point of view; however, this makes it difficult to increase components such as MS and TC, which increase more effectively through group dynamics, favoring resilience, cohesion among team members, and positive motivational climates through the feeling of belonging to the group [55,56]. The latter is an aspect of great value in adolescence [57], which is the stage at which the Junior category athletes are found.

However, no influence of age or rowing experience was demonstrated on the psychological components evaluated. To date, no longitudinal investigations have been carried out to establish causal relationships between both variables. In any case, it is necessary to consider the psychological variables at an early age, first to ensure and promote the health of athletes, and second, because they can act as predictors of the development of a successful sports career [58].

In this research, the results in the CPRD were not related to the results obtained in the championship. This lack of results is not consistent with previous studies that report an increase in sports performance in athletes with better psychological abilities [1,3,30,39,40,45,46,54]. However, this may also be due to the fact that the position achieved in the final classification was considered and no other variables related to performance, such as the times reached in the tests (variable not included as it differs according to the rowing modality in which the athletes participated) were considered.

On the other hand, this research provides evidence of the effectiveness of an intervention focused on CBT in youth rowers, using conventional psychological techniques with confirmed validity [29,51,59,60]. However, other types of programs may also be effective [3], such as Mindfulness–Acceptance–Commitment programs, which also lead to a reduction in anxiety, eating problems, and other psychological disorders; increase psychological flexibility; and also seem to improve sports performance [61].

The work of a sports psychologist in these youth ages is very relevant for the good sporting and social development of youth athletes, both in working with the athletes themselves and with coaches and parents [62,63]. These ages constitute a fundamental stage for the acquisition of good practices and habits for a future professional sports career or, simply, a healthy vital relationship with sport [44].

This research presents methodological limitations that must be recognized: the lack of a control group that confirms the efficacy of the applied intervention and the lack of a long-term follow-up that confirms the causal relationship between the different variables included. Another limitation was that, although the 10-week training camp was held at the Specialized High-Performance Rowing and Canoeing Centre, the athletes came from clubs all over Spain. Thus, the particularities of each club must be considered, since each of them has substantial differences in terms of organization, training of their coaches, and services they can provide to their athletes (physiotherapy, psychology, etc.). The use of a single assessment instrument limits the validity of the results and limits their generalizability, as it has not been possible to triangulate them with other measurement instruments. In order to guarantee the non-influence of extraneous variables such as gender, the relative effect of age, the small number of athletes, the brevity of the program, or those that may be generated by the psychologist who has intervened, it would have been necessary to demonstrate that these variables do not influence the result. In this sense, it would be very convenient to be able to develop randomized and controlled research. There was no follow-up after the last session of the intervention, so there is no data on whether these changes were maintained over time. In relation to the time of the season in which the intervention was carried out, it should be noted that it would be advisable to implement this type of program from the beginning of the season to the rest of the season. Finally, we would like to mention the need to continue studying the different theories and psychological models that may be involved in sports performance in young athletes with the aim of studying this subject in greater depth in order to increase their personal well-being, reduce early sporting abandonment, and prevent the development of psychological alterations that reduce their quality of life.

However, this study also has strengths that support the results obtained, such as the inclusion of a representative sample of the population group analyzed and the innovation that involves carrying out an intervention of this type in Junior category rowers, which are characteristics that have not been found in the literature to date. Furthermore, this study provides valuable information specific to the psychological characteristics of junior rowers.

## 5. Conclusions

The intervention based on mental training (CBT) in youth rowers in preparation for their participation in the Junior European Championship improved some of the CPRD variables. Specifically, it improved the SC and IPE, thus improving stress management in general and its relationship to performance evaluation. However, these improvements were not related to better results in the European Rowing Junior Championship.

Sports performance is determined by technical, tactical, physical, and psychological factors. Psychological training is very important for the improvement and/or maintenance of sports performance, producing very positive results in variables such as motivation, mental concentration, team cohesion, etc. with techniques such as visualization, goal setting, and relaxation through breathing, among others.

The present research provides evidence of the effectiveness of psychological interventions in the field of grassroots sports. Informing athletes from a young age about the benefits of sport psychology, showing them the variety of psychological characteristics that elite athletes possess and the available psychological skills that are commonly used to improve performance, can have great and positive consequences. In this sense, the figure of the sport psychologist is very relevant to carry out psychological interventions and thus stimulating the psychological baggage of athletes, also contextualized in young athletes, a stage of acquisition of good practices and habits for a future professional career as an athlete or, simply, a healthy life relationship with sport.

## Figures and Tables

**Figure 1 healthcare-10-02357-f001:**
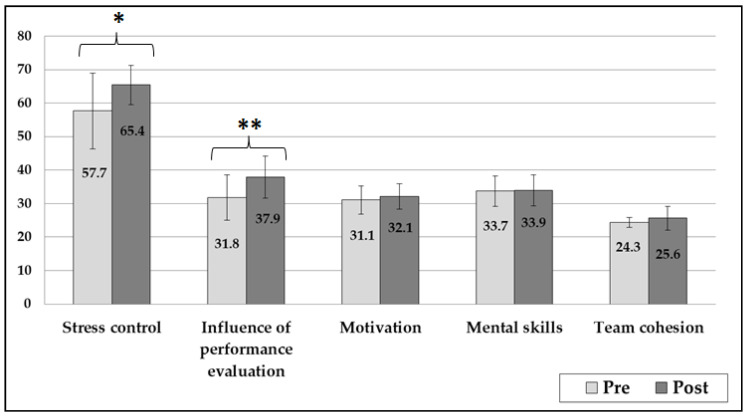
Changes in the psychological characteristics related to Sport Performance Questionnaire subscales (* *p* < 0.01; ** *p* < 0.001).

**Table 1 healthcare-10-02357-t001:** Characteristics of the participants at baseline.

Variable	All (*n* = 16)	Women (*n* = 7)	Men (*n* = 9)	*p* Value
Age (years)	17.1 ± 0.5	17.1 ± 0.4	17.1 ± 0.6	0.9
Experience (years)	6.2 ± 2.2	6.3 ± 2.4	6.1 ± 2.1	0.8
Psychological Characteristics Related to Sport Performance (points):				
Stress control	57.7 ± 11.3	61.4 ± 12.6	54.8 ± 6.3	0.01
Influence of performance evaluation	31.8 ± 6.8	33.1 ± 5	29.3 ± 7.8	0.29
Motivation	31.1 ± 4.2	31.4 ± 5.3	30.8 ± 3.3	0.77
Team cohesion	24.3 ± 1.5	24.9 ± 1.5	23.8 ± 1.6	0.8
Mental skills	33.7 ± 4.5	32.4 ± 4	34.7 ± 4.9	0.35
Championship table [*n* (%)]:				0.5
First	2 (12.5%)	0 (0%)	2 (22.2%)	
Second	2 (12.5%)	0 (0%)	2 (22.2%)	
Thrird	0 (0%)	0 (0%)	0 (0%)	
Fourth	7 (43.8%)	6 (85.7%)	1 (11.1%)	
Fith	0 (0%)	0 (0%)	0 (0%)	
Sixth	0 (0%)	0 (0%)	0 (0%)	
Seventh	4 (25%)	0 (0%)	4 (44.5%)	
All	15 (93.8%)	6 (85.7%)	9 (100%)	

**Table 2 healthcare-10-02357-t002:** Regression coefficients of the linear regression models for the variable championship (continuous variables) adjusted by years of rowing experience.

Variable	B	R^2^	SE	95% C.I.	ω^2^
Age	−2.58 *	0.36	1.02	0.38–4.78	0.25
Stress control	0.08	0.08	0.1	−0.15–0.3	−0.06
Influence of Performance Evaluation	0.08	0.05	0.1	−0.13–0.28	−0.02
Motivation	0.004	0.04	0.15	−0.32–0.33	−0.1
Team Cohesion	0.04	0.003	0.18	−0.34–0.41	−0.06
Mental Skills	0.11	0.0001	0.13	−0.28–0.3	−0.07

B: regression coefficient; R^2^: determination coefficient; SE: standard error; 95% C.I.: 95% confidence interval; ω^2^: omega squared. * *p* < 0.05.

## Data Availability

The data presented in this study are available upon request from the corresponding author.

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
