# Peer review of "Effects of a Cognitive-Behavioral Therapy Intervention on the Rowers of the Junior Spain National Team"

_healthcare, 2022, doi:10.3390/healthcare10122357_

Round 1
Reviewer 1 Report
General comments
The aim of the article is to study the effects of a CBT intervention on different psychological aspects in a junior team of rowers (n=16). To do so, authors carry out a 10-week treatment, assessing with the CPRD before and after the treatment and then looking at the result in a championship. The results showed improvement in two scales (stress control and influence of performance evaluation). Afterwards some different results are shown according to gender. However, these results had no effect on the outcome of the competition.
Having reviewed the article in depth, I consider it to be a very interesting article that addresses a necessary aspect of sport performance, specifically applying pre-competition intervention to try to predict or improve the outcome of competition. Furthermore, it is studied in a major competition, which reinforces the importance of the paper. However, despite the effort and the proposal of the work, I consider that there are aspects that should be improved.
The main problem I consider is that it is a study that applies a therapy without having described a problem before. That is to say, the objective of the treatment is not clear and therefore it is not justified. Were there deficits in the variables evaluated in the athletes? Furthermore, the treatment given (according to table 1) is general and does not directly target the variables that are measured afterwards (e.g. pre-competitive stress and motivation are not treated). If these aspects are not addressed, no direct change in these variables can be expected. Furthermore, it is not clarified whether the values in the questionnaire differ from other populations or previous measurements.
In short, I detect many aspects that do not justify the research carried out. From my point of view, despite the merit of the article, it should be reformulated in order to justify the hypotheses and the work carried out, highlighting, of course, the limitations of the study.
Specific comments
Abstract
Abstract should be reformulated, focusing on the introduction part in CBT and gender differences, not in general interventions. Moreover, abstract needs more information about the method.
Introduction/objective
From point of view, it is not justified what this type of therapy can bring in comparison to other treatments. CBT is broad and there are many techniques that can be applied and as explained above, there is no justification of what has been worked on in terms of the objectives. In fact, the introduction is very diffuse, and the different treatments are not explained in comparison to the one that has been used and the measures that are evaluated are not justified. This section should be reworded to allow for justification of the study
The issue of gender is not justified in the introduction and if it is an objective, it should be dealt with in depth in the introduction. In fact, the objectives should be well justified and made explicit so that the results can respond to them. Moreover, justifying the objective of the study “because it is considered necessary” is not a scientific reason.
Method
A description of the larger sample would be needed. It is difficult to know what kind of athletes we are talking about. Moreover, it is stated that the starting point is 27 rowers, but there is no explanation as to why there were only 16 in the end.
The intervention programme needs to be explained in more depth so that other researchers could replicate the study. The information presented is minimal and not very specific.
Results
The results are not adjusted to the objectives.
Do not use the word “sexes” but “gender”.
Discussion
In the discussion, the results are not sufficiently compared with other populations or other types of treatments. It should be made clear what these results contribute with respect to other treatments in terms of the variables evaluated. Furthermore, the issue of age and experience is not relevant in a group of athletes with a very specific age range. Therefore, it should not be discussed. In addition, the gender results should be addressed more deeply, as in the introduction. The discussion should be redrawn with more detail and a more applied perspective.
Author Response
Dear Editor and Reviewers of Healthcare:
Thank you very much for your suggestions and contributions to improve the quality of the manuscript. Following your indications, we respond, point by point, to the reviewers' comments.
In the text, all the modified or added sentences have been written in red to facilitate the correction by the reviewers.
Reviewer #1:
- Having reviewed the article in depth, I consider it to be a very interesting article that addresses a necessary aspect of sport performance, specifically applying pre-competition intervention to try to predict or improve the outcome of competition. Furthermore, it is studied in a major competition, which reinforces the importance of the paper. However, despite the effort and the proposal of the work, I consider that there are aspects that should be improved.
The authors are very grateful for the reviewer's feedback.
- The main problem I consider is that it is a study that applies a therapy without having described a problem before. That is to say, the objective of the treatment is not clear and therefore it is not justified. Were there deficits in the variables evaluated in the athletes? Furthermore, the treatment given (according to table 1) is general and does not directly target the variables that are measured afterwards (e.g. pre-competitive stress and motivation are not treated). If these aspects are not addressed, no direct change in these variables can be expected. Furthermore, it is not clarified whether the values in the questionnaire differ from other populations or previous measurements.
In short, I detect many aspects that do not justify the research carried out. From my point of view, despite the merit of the article, it should be reformulated in order to justify the hypotheses and the work carried out, highlighting, of course, the limitations of the study.
Thank you very much for your advice.
We have modified the introduction and it is now clearer to the reader that this is a Cognitive-Behavioral Therapy-based intervention programme to analyse and improve as far as possible their sporting performance.
A section on the limitations of the study has been introduced.
- Abstract: Abstract should be reformulated, focusing on the introduction part in CBT and gender differences, not in general interventions. Moreover, abstract needs more information about the method.
The Abstract has been rewritten following your advice and in line with the modifications made to the main text of the manuscript.
- Introduction/objective: From point of view, it is not justified what this type of therapy can bring in comparison to other treatments. CBT is broad and there are many techniques that can be applied and as explained above, there is no justification of what has been worked on in terms of the objectives. In fact, the introduction is very diffuse, and the different treatments are not explained in comparison to the one that has been used and the measures that are evaluated are not justified. This section should be reworded to allow for justification of the study.
The introduction has been rewritten and new information has been introduced on the different psychological intervention techniques and programs used and why we have opted for Cognitive-Behavioral Therapy.
- Introduction/objective: The issue of gender is not justified in the introduction and if it is an objective, it should be dealt with in depth in the introduction. In fact, the objectives should be well justified and made explicit so that the results can respond to them. Moreover, justifying the objective of the study “because it is considered necessary” is not a scientific reason.
A section on gender has been introduced and added to the objective of the study.
- Method: A description of the larger sample would be needed. It is difficult to know what kind of athletes we are talking about. Moreover, it is stated that the starting point is 27 rowers, but there is no explanation as to why there were only 16 in the end.
The authors have included the reasons for non-participation of team members not included in the research.
- Method: The intervention programme needs to be explained in more depth so that other researchers could replicate the study. The information presented is minimal and not very specific.The authors have substituted as many references older than ten years as possible (ensuring the quality and accuracy of the content of the works cited).
The authors have added a detailed description of the content developed in the different sessions of the intervention and the materials and methods used. This work can now be replicated by any researcher.
- Results: The results are not adjusted to the objectives.
The authors understand the reviewer's comment and greatly regret having made this formal error in scientific writing.
The authors have rewritten and reworded the research objectives to make the main text consistent across the different sections.
- Results: Do not use the word “sexes” but “gender”.
The authors greatly regret having made this very important mistake. The term has been corrected throughout the text.
- Discussion: In the discussion, the results are not sufficiently compared with other populations or other types of treatments. It should be made clear what these results contribute with respect to other treatments in terms of the variables evaluated. Furthermore, the issue of age and experience is not relevant in a group of athletes with a very specific age range. Therefore, it should not be discussed. In addition, the gender results should be addressed more deeply, as in the introduction. The discussion should be redrawn with more detail and a more applied perspective.
The authors have written a paragraph on the interpretation of the results according to gender.
In addition, we have removed some sentences that were not fully supported by the evidence.
Once again, thank you very much for the time spent and the interest shown in this work; as well as in the positive evaluations you have given of it.
Receive a warm greeting,
The authors.
Reviewer 2 Report
Thank you for the opportunity to review this manuscript. Unfortunately, I have significant concerns regarding the conceptual basis for the research, coherence of approach and the discussion of findings.
Introduction
Greater conceptual clarity required, you mention motivation and confidence as being cognitive skills, are they skills or outcomes (cf. Collins et al., 2018)? Please highlight conceptual framework used and clarify differences between psychological skills, cognitive skills and other terms used. This is critical for the reader to evaluate the level and nature of your intervention.
Please further outline the evidence supporting the efficacy and use of CBT in psychology. As it stands, this section is significantly under weighted and requires greater elaboration for the reader to understand the logic of the intervention. Please supply evidence supporting CBT’s efficacy for managing arousal, concentration, self-confidence, pre competition routines and transfer to wider life - or, for the skills that might underpin these outcomes? In addition, if these were the aims for your intervention, why did you choose to use the Psychological Characteristics Related to Sport Performance Questionnaire instead of other questionnaires, given that is seems to focus on stress control (which is a different construct to arousal control), performance evaluation, motivation and team cohesion?
There is no literature on transfer of psychological skills in the introduction, this seems like a critical element of your investigation?
Materials and Methods
How were the convenience sample selected? Were they those eventually selected? If so, were the authors aware of the team selection prior to the training camp? Did selection have any implications for the data collection process? How were athletes informed of the data collection - please outline measures taken to ensure it didn’t interfere with quality of data via self presentation bias.
Please supply more information regarding the nature of the methods (e.g., daily exercises) and how the psychologist integrated their work with coaches and training. This would seem critical given the low efficacy of workshop based methods for transfer to actual performance.
How do the methods used correspond with an individualised CBT approach given that they appear to be delivered in the group setting?
Were any means of analysis used to triangulate questionnaire data? If not, this seems to be a critical limitation that needs to be discussed, in addition, it would be appropriate to discuss the limitations of questionnaire data as it is.
Specifics
L16 - missing full stop
L20 - unsure what the ‘concentration’ is? Similar throughout paper
L25 - Female athletes may be a more appropriate term
L37 - Please clarify this sentence
L43 - Please clarify the increased importance of wellbeing for adolescent athletes
L44 - Statement made does not follow from previous points. It is unclear why the work of a psychologist is essential, or why individualised interventions are necessary. This needs significant qualifying or rephrasing.
L77 - check wording “the athletes were concentrated”
L82 - do you mean arousal?
Results
L132 - Please clarify meaning “position they reached in the Championship” - were they all taking part in individual events?
It would seem face valid that if you did use CBT that it would improve those elements of performance like stress control rather than others that you measured. Why did you choose to measure constructs that do not seem to be related to CBT?
Discussion
There are a number of major weaknesses in the discussion which critically limit the paper overall. Some of which are outlined below:
L171 - You mention that athletes were overwhelmed throughout the season, what data did you capture to support this statement?
L172 - You state that this stress has a negative impact on academic-sport compatibility. Please clarify the wording and point to the elements of your study that support this statement
L176 - This seems an overly generalised statement about sex differences, I am not sure the data you have supports this hypothesis
L197 - Sorry, this statement does not follow, nor is appropriate unless you captured data comparing with the skill base of other competitiors at the championships. Similarly, are you referring to short, or long term performance? Either way, this statement seems to directly contradict the point you make about performance relying on a variety of variables (L33)
L203 - Seems unrelated to the study? Do you have data showing that older rowers were heavier? If not, I suggest deleting
L209 - I am unclear how the rowers being in different clubs impacts the nature of the intervention. If this is because they weren’t in a training camp for 10 weeks, please outline this in the intervention with a far clearer view of the nature of the intervention.
L215 - Unclear on meaning: “inclusion of the total population universe”
Collins, D., MacNamara, Á., & Cruickshank, A. (2019). Research and Practice in Talent Identification and Development—Some Thoughts on the State of Play. Journal of Applied Sport Psychology, 31(3), 340-351. https://doi.org/10.1080/10413200.2018.1475430
Author Response
Dear Editor and Reviewers of Healthcare:
Thank you very much for your suggestions and contributions to improve the quality of the manuscript. Following your indications, we respond, point by point, to the reviewers' comments.
In the text, all the modified or added sentences have been written in red to facilitate the correction by the reviewers.
Reviewer #2:
- Thank you for the opportunity to review this manuscript. Unfortunately, I have significant concerns regarding the conceptual basis for the research, coherence of approach and the discussion of findings.
The authors would like to thank you for your comments for improvement. Moreover, now, after all the improvements and corrections made, we hope that your concerns will be resolved.
- Introduction: Greater conceptual clarity required, you mention motivation and confidence as being cognitive skills, are they skills or outcomes (cf. Collins et al., 2018)? Please highlight conceptual framework used and clarify differences between psychological skills, cognitive skills and other terms used. This is critical for the reader to evaluate the level and nature of your intervention.
The introduction has been substantially modified. We have opted for CBT because it is the technique used in most psychological intervention programs to improve sports performance with young athletes.
- Introduction: Please further outline the evidence supporting the efficacy and use of CBT in psychology. As it stands, this section is significantly under weighted and requires greater elaboration for the reader to understand the logic of the intervention. Please supply evidence supporting CBT’s efficacy for managing arousal, concentration, self-confidence, pre competition routines and transfer to wider life - or, for the skills that might underpin these outcomes?
The introduction has been substantially modified and the importance of Cognitive-Behavioral Therapy over other psychological techniques is now stated. It is accompanied by an updated bibliography.
- Introduction: In addition, if these were the aims for your intervention, why did you choose to use the Psychological Characteristics Related to Sport Performance Questionnaire instead of other questionnaires, given that is seems to focus on stress control (which is a different construct to arousal control), performance evaluation, motivation and team cohesion?
We have opted for the Psychological Characteristics Related to Sport Performance Questionnaire because of its fabulous validity and reliability in the Spanish population. In Spain, it is the most widely used instrument and has the best correlation with sports performance.
- Introduction: There is no literature on transfer of psychological skills in the introduction, this seems like a critical element of your investigation?
Literature relating the presence of psychological skills and sport performance is referenced now in this section.
- Materials and Methods: How were the convenience sample selected? Were they those eventually selected? If so, were the authors aware of the team selection prior to the training camp? Did selection have any implications for the data collection process? How were athletes informed of the data collection - please outline measures taken to ensure it didn’t interfere with quality of data via self presentation bias.
The participant selection process and reasons for non-participation of team members have been added in the manuscript.
- Materials and Methods: Please supply more information regarding the nature of the methods (e.g., daily exercises) and how the psychologist integrated their work with coaches and training. This would seem critical given the low efficacy of workshop based methods for transfer to actual performance.
The authors have expanded the description of the intervention applied. In addition, we have better described the materials and methods used.
- Materials and Methods: How do the methods used correspond with an individualised CBT approach given that they appear to be delivered in the group setting?
Since in the group sessions there was total freedom on the part of the athletes to actively participate and share their opinions or experiences, this allowed the psychologist to address specific needs of those participants who expressed particular concerns or needs.
- Materials and Methods: Were any means of analysis used to triangulate questionnaire data? If not, this seems to be a critical limitation that needs to be discussed, in addition, it would be appropriate to discuss the limitations of questionnaire data as it is.
The authors have expanded the limitations of the research to include this important methodological detail.
- L16 - Missing full stop.
The spelling mark has been added. The authors are very sorry to make such mistakes.
- L20 - Unsure what the ‘concentration’ is? Similar throughout paper.
The authors have replaced this term with "mental concentration".
- L25 - Female athletes may be a more appropriate term.
The authors have replaced the term following your advice.
- L37 - Please clarify this sentence.
The authors have corrected the sentence to make it clearer for interpretation.
- L43 - Please clarify the increased importance of wellbeing for adolescent athletes.
Clearly, we mean young people, like those in our sample.
- L44 - Statement made does not follow from previous points. It is unclear why the work of a psychologist is essential, or why individualised interventions are necessary. This needs significant qualifying or rephrasing.
With the modification of the Introduction, we believe that it shows evidence of the importance of the implementation of this program with young athletes.
- L77 - Check wording “the athletes were concentrated”.
Revised and modified.
- L82 - Do you mean arousal?
Yes, the authors regret this translation error. The entire manuscript has been revised by a native English translator.
- Results: L132 - Please clarify meaning “position they reached in the Championship” - were they all taking part in individual events.
Yes, all research participants were participants in at least one individual test and their best achieved position was used as a study variable.
This detail has been included in the Material and Methods section.
- Results: It would seem face valid that if you did use CBT that it would improve those elements of performance like stress control rather than others that you measured. Why did you choose to measure constructs that do not seem to be related to CBT?
We have used the Psychological Characteristics Related to Sport Performance Questionnaire as a measuring instrument. We did not want to use more scales to avoid athlete fatigue when covering multiple questionnaires.
- Discussion: There are a number of major weaknesses in the discussion which critically limit the paper overall. Some of which are outlined below: L171 - You mention that athletes were overwhelmed throughout the season, what data did you capture to support this statement?
That sentence has been removed from the manuscript because it represents a claim that researchers are not entitled to make. We regret the error.
- Discussion: L172 - You state that this stress has a negative impact on academic-sport compatibility. Please clarify the wording and point to the elements of your study that support this statement.
That sentence has also been removed from the manuscript because it represents a claim that researchers are not entitled to make. We regret the error.
- Discussion: L176 - This seems an overly generalised statement about sex differences, I am not sure the data you have supports this hypothesis.
The authors have limited the statement to female athletes. In addition, we provide two references that support this explanation.
However, we have rewritten the sentence so as not to fall into lucubration.
- Discussion: L197 - Sorry, this statement does not follow, nor is appropriate unless you captured data comparing with the skill base of other competitiors at the championships. Similarly, are you referring to short, or long term performance? Either way, this statement seems to directly contradict the point you make about performance relying on a variety of variables (L33).
The authors have limited this statement to the results obtained in this research.
- Discussion: L203 - Seems unrelated to the study? Do you have data showing that older rowers were heavier? If not, I suggest deleting.
That sentence has also been removed from the manuscript because it represents a claim that researchers are not entitled to make. We regret the error.
- Discussion: L209 - I am unclear how the rowers being in different clubs impacts the nature of the intervention. If this is because they weren’t in a training camp for 10 weeks, please outline this in the intervention with a far clearer view of the nature of the intervention.
It has been clarified that the athletes come from clubs all over Spain but the 10-week training camp was held at the center specialized in rowing and canoeing. The work carried out in the 10 sessions was also explained in more detail.
- L215 - Unclear on meaning: “inclusion of the total population universe”.
The authors have rephrased the sentence for easier interpretation.
Once again, thank you very much for the time spent and the interest shown in this work; as well as in the positive evaluations you have given of it.
Receive a warm greeting,
The authors.
Reviewer 3 Report
The intetion of this study is interesting and may contribute to young atheletes.
However, the statiscal methods and results presentation section must be improved.
Below are some suggestions:
Abstract: please provide practical implications upon this research.
P. 4 Ln #126, usually, F test was applied to test the significance of regressors instead of R-square.
P5. Ln #141-143, there is no table to indicate your descriptions and what do you mean by "men maintained the same results in the SC and IPE suscales......."?
P5. Ln #144-146, please provide more details. For example, whic sex was higeher? May a table will be nice.
P6. Ln #151-153 I don't undertand the analyses. Why performed the test? What does position indicating in this research? Besides, there were discussions related to this result.
P6. Ln #170-172 Based on what ground you brought about this conclusion?
P6. Ln #173-174 This discussion is confusing. I can't related this with the results from Ln #141-143.
To sum up, there are two main suggestions to this article:
1. Make the results presnetation clearer so the readers can easily comprehend your work.
2. Please provide practical implicatons in the conclusion.
Wish you the best!
Author Response
Dear Editor and Reviewers of Healthcare:
Thank you very much for your suggestions and contributions to improve the quality of the manuscript. Following your indications, we respond, point by point, to the reviewers' comments.
In the text, all the modified or added sentences have been written in red to facilitate the correction by the reviewers.
Reviewer #3:
- The intetion of this study is interesting and may contribute to young atheletes. However, the statiscal methods and results presentation section must be improved.
Thank you very much for your feedback for improvement.
- Abstract: please provide practical implications upon this research.
The Abstract has been rewritten following your advice and in line with the modifications made to the main text of the manuscript.
- P5. Ln #141-143, there is no table to indicate your descriptions and what do you mean by "men maintained the same results in the SC and IPE suscales......."?
The authors have chosen to represent the figures in the form of a Figure (in addition, the numerical results for each variable and study subgroup are also shown in the Figure).
- P6. Ln #170-172 Based on what ground you brought about this conclusion?
That sentence has been removed from the manuscript because it represents a claim that researchers are not entitled to make. We regret the error.
- P6. Ln #173-174 This discussion is confusing. I can't related this with the results from Ln #141-143.
The authors have written a paragraph on the interpretation of the results according to gender.
In addition, we have removed some sentences that were not fully supported by the evidence.
- To sum up, there are two main suggestions to this article: (a) Make the results presnetation clearer so the readers can easily comprehend your work.
The authors have modified the manuscript: we have not changed much of the results but we have changed all the other sections so that all the sections of the main text are consistent with each other.
- (b) Please provide practical implicatons in the conclusion.
The authors have expanded the Conclusions on your advice.
Once again, thank you very much for the time spent and the interest shown in this work; as well as in the positive evaluations you have given of it.
Receive a warm greeting,
The authors.
Round 2
Reviewer 1 Report
Thank you for revising your manuscript including my suggestion.
Reviewer 3 Report
Authors had revised article accordingly to reviewers' comment.